# The Effect of Omega-3 Fatty Acid Supplementation on Serum Adipocytokines, Lipid Profile and Biochemical Markers of Inflammation in Recreational Runners

**DOI:** 10.3390/nu13020456

**Published:** 2021-01-29

**Authors:** Aleksandra Żebrowska, Barbara Hall, Anna Stolecka-Warzecha, Arkadiusz Stanula, Ewa Sadowska-Krępa

**Affiliations:** 1Institute of Sport Sciences, Academy of Physical Education in Katowice, Mikołowska Street 72a, 40-065 Katowice, Poland; a.stanula@awf.katowice.pl (A.S.); e.sadowska-krepa@awf.katowice.pl (E.S.-K.); 2School of Biological Sciences, The University of Manchester, Manchester M13 9PL, UK; barbara.hall@manchester.ac.uk; 3Department of Basic Biomedical Sciences, Silesia Medical University, 40-055 Katowice, Poland; astolecka@sum.edu.pl

**Keywords:** omega-3 index, diet, adipocytokines, inflammation, physical exercise

## Abstract

Background: The study aimed to evaluate the effects of a 3-week ω-3 PUFA supplementation on serum adipocytokines (i.e., adiponectin, leptin), neuregulin-4 (NRG4) and erythrocyte omega-3 (ω-3) fatty acid content, as well as the blood antioxidant defense capacity in non-elite endurance runners. Methods: Twenty-four runners were randomized into two groups: the supplemented group, who received omega free fatty acids extract containing 142 mg of EPA, 267 mg of DHA, 12 mg of vitamin E and 5 µg of vitamin D, each administrated at a dose of six capsules twice a day for three weeks, or the placebo group. Venous blood samples were withdrawn at the start and at the end of the study protocols to estimate serum biochemical variables. Results: A significantly higher ω-3 index and lower AA/EPA ratio was observed after ω-3 PUFA compared to pre-supplementation levels (*p* < 0.001 and *p* < 0.001, respectively). An increase in baseline adiponectin and NRG4 levels, as well as a decrease of leptin concentration and lipid profile improvement, were observed in subjects after a ω-3 PUFA diet. The increased ω-3 index had a significant effect on TNFα levels and a serum marker of antioxidant defense. Conclusions: The ω-3 PUFA extract with added vitamin E and D supplementation may have a positive effect on the function of the adipocyte tissue, as well as the ability to prevent cardiovascular complications in athletes.

## 1. Introduction

A diet rich in omega-3 polyunsaturated fatty acids (ω-3 PUFA), such as docosahexaenoic acid (DHA), and eicosapentaeonic acid (EPA), has been shown to have a positive effect on human health [1,2,3]. Numerous studies demonstrated various benefits of ω-3 PUFA consumption, including the prevention of hyperlipidemia [4,5,6] and the development of cardiovascular disease (CVD) [7,8], which are both highly prevalent in Western populations [9,10,11]. It is now well-established that inflammation is a potent promoter of CVD [12]. Acute inflammation is a physiological defense mechanism which helps to restore homeostasis at the site of infection or damaged tissue [13,14]. Regulated inflammation is essential for maintaining health, although excessive and lasting inflammatory processes will lead to tissue damage and disease [14,15]. The health benefits of ω-3 PUFA can be attributed to their role in reducing the production of inflammatory molecules and reactive oxygen species (ROS) [13]. EPA and DHA have been shown to inhibit the synthesis of arachidonic acid (AA)-derived eicosanoids, which are key mediators and regulators of inflammation and immunity [14]. The eicosanoids made from ω-6 PUFA such as AA have been suggested as more potent mediators of inflammation than those made from ω-3 PUFA. Thus, higher concentrations of EPA and DHA than AA tip the eicosanoid balance toward less inflammatory activity. Combining an ω-3 PUFA diet with vitamin D and antioxidants (vitamin E) has been suggested to increase the anti-inflammatory processes and therapeutic effects of ω-3 fatty acids [15,16]. A diet rich in EPA and DHA alternates the biological membranes lipid composition, modulating their physical properties, such as fluidity and permeability, positively influencing a myriad of physiological processes [17,18]. Moreover, ω-3 PUFA positively affects lipid homeostasis by modifying the adipose tissue endocrine and paracrine functions, and stimulating lipid oxidation [4,10]. 

Moderate intensity exercise training has various health benefits, and is recommended for the prevention of metabolic and cardiovascular diseases [19,20]. Although much data pointed out the exercise-induced physiological adaptations, including improved blood lipid and blood pressure profiles, as well as enhanced physical performance, an increased risk of exercise-induced myocardial infarction has also been reported [21,22,23]. Among the potential mechanisms triggering this exercise-related coronary syndrome are increased release of proinflammatory mediators and an imbalance between the anti-oxidative and oxidative effects of exercise [21,24,25]. It has been suggested that some of these unfavorable changes may be related to the low supply of ω-3 fatty acids [3,26]. Previous studies indicated that prolonged endurance exercise may negatively influence the ω-3 PUFA profile, as well as their ability to reduce inflammation [27,28,29].

It has been shown that dietary supplementation with ω-3 PUFA induces the expression of endogenous antioxidant genes, such as superoxide dismutase (SOD), glutathione peroxidase (GPX) and catalase (CAT), as well as anti-inflammatory cytokines in animal models [30], healthy individuals [31] and athletes participating in high-intensity endurance exercise [32]. The anti-inflammatory and anti-oxidant properties of ω-3 PUFA can be also supported by their ability to decrease the secretion of proinflammatory cytokines, such as tumor necrosis factor alpha (TNFα) and interleukin 6 (IL-6), as well as ROS via modulating the function of the neutrophils [2].

Effects of ω-3 fatty acids on the secretion of adipocytokines which exert both immunomodulatory and metabolic effects, have not been extensively investigated yet [14,15,33]. Adiponectin is an adipocytokine which increases insulin sensitivity augmenting glucose uptake, and promotes fatty acids oxidation in the skeletal muscle tissue [34]. It also plays a key regulatory role in inhibiting gluconeogenesis, and therefore maintaining normoglycaemia [35,36]. Adiponectin has also been shown to inhibit inflammation by blocking NF-_K_B activation and reducing levels of pro-inflammatory cytokines TNFα, IL-6 and IL-18 [37,38]. It has been demonstrated that adiponectin release increases in response to EPA and DHA supplementation [10]. 

Evidence also suggests a negative, dose-dependent relationship between ω-3 PUFA intake and circulating levels of leptin in non-obese subjects [5,10]. Nevertheless, in obese individuals, a hyper energetic diet with a high ω-3 PUFA consumption may increase levels of this adipocytokine [39]. Elevated leptin has been shown to induce leptin resistance, leading to the development of metabolic disorders, such as insulin resistance and dyslipidemia [40], as well as CVD [41]. Hyperleptinaemia has been also shown to alter cytokine signaling, including increased production of a pro-inflammatory cytokine TNFα [10,42].

Neuregulin-4 (NRG-4), a member of the neuregulin family (NRG1–NRG4), has recently been introduced as a novel adipocytokine with beneficial metabolic effect [43,44,45,46]. NRG-4 produces signals through the receptor tyrosine—protein kinases/human epidermal growth factor receptors (ErB/HER) and exerts pleiotropic effects that ultimately regulate energy metabolism and insulin sensitivity, as well as protects against oxidative stress [47]. It has been demonstrated that higher NRG-4 levels are associated with improvement in glucose homeostasis, maintenance of lipid balance, prevention of insulin resistance and weight gain [48,49]. Recently, NRG-4 has been shown to directly reduce lipogenesis in hepatocytes and activate brown adipose tissue (BAT) to reduce diet-induced obesity [50,51]. It is possible that NRG-4 pathway may be stimulated by the nutritional intervention and modification of dietary lipid consumption [43,52]. Significant increases in adiponectin and a reduction in leptin levels following ω-3 supplementation have been reported previously [10,33], although data on effects of ω-3 on circulating NRG-4, lipid profile and anti-oxidant defense are still sparse.

The ω-3 index defined as the sum of EPA + DHA as % of total fatty acids in erythrocyte membrane is a measurement of intake and tissue levels of EPA plus DHA [1,53,54]. It has been confirmed that a diet rich in ω-3 fatty acids increases the ω-3 index, therefore reducing risk for cardiovascular events in general and athletic population [55]. The ω-3 index is now commonly used in clinical laboratories to identify a suboptimal intake of ω-3 PUFA, and risk for a sudden myocardial infarction after strenuous exercise [32,56,57]. 

Since competitive and recreational athletes are considered to be at higher risk of myocardial infarction resulting from excessive or long-lasting inflammation, the main objective of the study was to investigate the effects of ω-3 PUFA supplementation on the ω-3 index, serum levels of adipocytokines and the blood antioxidant defense system in a group of healthy recreational runners. 

## 2. Material and Methods

### 2.1. Subjects

Twenty-four healthy, physically active male volunteers who had no history of diseases and medication and/or no infection prior to and at the time of the study were included in this blind, randomized placebo-controlled trial, and assigned to either the placebo or the ω-3 PUFA group. The study participants were recreational marathon runners (the marathon distance = 42.195 km or 26.2 miles) recruited through an online running academic website. The inclusion criteria were as follows: completion of at least one marathon in the last twelve months; written informed consent to take part in the study; no acute metabolic and cardiovascular diseases; or no infection prior to the study. The exclusion criteria used in order to eliminate factors which might influence the biochemical parameters were as follows: evidence of metabolic and hemodynamic dysfunction, inflammatory diseases in the preceding three months, and cigarette smoking. All subjects reported that they were not taking fish oil supplements or any medication that could affect their cardiovascular function. For the entire duration of the experiment, the training regimen (intensity, duration and frequency of training) did not differ between the subjects. They were instructed to abstain from exercise within 24 h before the biochemical measurements.

Somatic characteristics of the participants and their training status are presented in Table 1. Participants’ body composition was evaluated using a model In Body220 analyser (Biospace Inc., Seoul, Korea). At the baseline, before the treatment protocol, heart rate (HR) (PE-3000 Sport-Tester, Polar Inc., Kempele, Finland) and systolic and diastolic blood pressure (SBP/DBP) (HEM-907 XL, Omron Corporation, Kyoto, Japan) were measured in all subjects. The graded treadmill exercise test (HP/Cosmos-Pulsar, Munich, Germany) was performed to measure individual VO_2_max (Matalyzer 3B, Cortex, Leipzig, Germany). The subjects’ training volume was calculated and presented in Table 1. Their marathon finishing times varied between 3 h 23 min and 4 h 18 min.

Three weeks prior to the study, all participants were put on a mixed diet (2400 ± 220 kcal/day). The composition of the diet of each study subject was calculated with a dedicated software (Dietus, B.U.I. InFit. Warsaw, Poland). Mean dietary energy supply and mean daily fat, carbohydrate, protein, vitamins and fatty acid intakes were compared between the groups (Table 2). Then, the diet regimen was continued with either the ω-3 PUFA or placebo administration for three weeks. None of the respondents consumed fish or other oils that would additionally modify a dietary fatty acids intake. Subjects were asked to follow the recommended diet, and compliance with it was evaluated during obligatory weekly visits in the laboratory. The subjects were then asked whether they had consumed all the recommended foods, and if any additional foods had been eaten.

The biochemical variables were analyzed before and after each diet protocol. 

All individuals were informed about the aim of the research, the possibility of refusal of the participation, and provided written informed consent. The study was approved by the Local Ethics Committee (Ethics Committee decision KBN 3.2016), and conformed to the standards set by the Declaration of Helsinki. 

### 2.2. Supplementation Procedure 

The participants were randomly allocated to two groups: (1) the control placebo group (placebo, *n* = 12); and (2) the study group (ω-3 PUFA, *n* = 12) supplemented with standardized omega free fatty acids extract, in the form of soft gelatinous capsules (Bergen OMEGA-3 Natural Pharmaceutic, Warsaw, Poland) administrated at a dose of six capsules per day, four in the morning and two in the evening, 30 min after a meal, for three weeks. Each capsule consisted of concentrated and refined sardine and anchovy oil containing 90% of ω-3 PUFA (142 mg of EPA, 267 mg of DHA), 12 mg of vitamin E and 5 µg of vitamin D. Daily, each subject from the ω-3 PUFA group was supplemented with 852 mg EPA, 1602 mg DHA, and 72 mg and 30 µg of vitamin E and D, respectively. The control group consumed an equivalent number of placebo gelatin capsules containing microcrystalline cellulose, magnesium stearate and lactose monohydrate instead of fish extract (Natural Pharmaceutic, Warsaw, Poland).

### 2.3. Biochemical Analyses

All clinical data were obtained after an overnight fast. All subjects reported to the laboratory and had venous blood drawn for the determination of adiponectin, leptin, Neuregulin-4, pro-inflammatory cytokines, blood antioxidant status and lipid profile. The blood samples were collected to determine the aforementioned markers before the study protocols (pre-supplementation vs. pre-placebo) and immediately after the diet intervention (i.e., post-supplementation or post-placebo protocol).

Blood samples from the antecubital vein were collected and were centrifuged for 15 min at 1000× *g*. The obtained serum was aliquoted and kept frozen at −80 °C (for a period no longer than eight months, without repeated freezing) until analyses of adiponectin, leptin, NRG-4, α-tocopherol, TNF-α, Il-6 and activities of antioxidant enzymes, i.e., SOD, GPX, CAT and reduced glutathione (GR). Fresh whole blood samples were immediately assayed for reduced glutathione (GSH) by a calorimetric method [58].

To quantify the percentage of PUFA in the erythrocyte cell membranes, the whole blood spot sample was discarded, and a drop was then placed onto a spot card pretreated with an antioxidant blend to prevent oxidative loss of PUFA. Blood spot cards were dried at room temperature and subsequently stored in a −80 °C freezer until shipment. Omega 3 fatty acid index (HS-Omega-3 Index^®^) with determination of the ratio of saturated fatty acids to monounsaturated fatty acids (SFA/MUFA), the TRANS fat index and the ratio of arachidonic acid to eicosapentaenoic acid (AA/EPA) were determined using GC-FID Gas Chromatography-Flame Ionization Detector. The omega 3 index is defined as the sum of EPA and DHA expressed as a percentage of total weight of fatty acids in erythrocyte membrane (Appendix A) [9]. It is measured in dried blood spots (DBS) and is a qualitative (%) not quantitative (µg/mL) expression of total fatty acids [9]. DBS analysis is a repeatable measurement which is relatively non-invasive. The analyses were performed in the certified laboratory (Centro Diagnostico Delta, Apollosa, Italy; certificate No. 8449/2010).

The serum adiponectin levels were measured using radioimmunoassay Adiponectin RIA KIT (ACRP30, EMD Millipore Co., Billerica, MA, USA). The intra-assay and inter-assay coefficients of variation values were <6.0% and 7.0%, respectively. Concentrations of serum leptin were measured using Human Leptin ELISA kit (DRG International, Inc., Springfield, NY, USA, AM). Intra- and inter-assay coefficients of variation for leptin were <7.0% and <9.0%, respectively. Serum levels of NRG4 and α-tocopherol were measured by an enzyme-linked immunosorbent assay ELISA kit (Bioassay Technology Laboratory, China). Serum TNF-α levels were measured by immunoassays kit (DIAsource, Ottignies-Louvain-la-Neuve, Belgium). Intra- and inter-assay coefficients of variation for NRG4 were <8.0% and <10.0%, respectively. Serum levels of IL-6 were measured using Human IL-6 High Sensitive ELISA kit (Diacone, Besançon, France). Intra- and inter-assay coefficients of variation for IL-6 were <4.0% and <6.0%, respectively, with a sensitivity of 0.7 pg/mL.

The activity of RBC antioxidant enzymes, i.e., SOD (EC 1.15.1.1), GPX (EC 1.11.1.9), CAT (EC 1.11.1.6), GR (EC 1.6.4.2) was analyzed as previously described [59]. Plasma lipid peroxidation product (malondialdehyde: MDA) level were assessed using the thiobarbituric acid (TBARS, MDA) reaction according to Buege and Aust [60]. Blood glucose (BG) concentration was measured by enzymatic method (glucose dehydrogenase) (Glucose 201+, HemoCue, Ängelholm, Sweden). 

Concentrations of serum total cholesterol (TCh), triglycerides (TG), low-density lipoprotein cholesterol (LDL-C), and high-density lipoprotein cholesterol (HDL-C) were assayed using enzymatic methods and a clinical chemistry analyzer (RA-XT, Technicon Instruments Corporation, Tarrytown, NY, USA, AM). The intra-assay CV for these assays was below 5.0%.

Biochemical analyses were performed in our certified laboratory fulfilling the requirements of PN EN-ISO 9001:2009 (certificate No. 129/2015). All biochemical analyses were performed according to the instructions provided by the manufacturers of the laboratory tests used in this study.

### 2.4. Statistical Analysis

All analyses were performed using the Statistica v. 12 statistical software package (StatSoft, Tulsa, OK, USA) and the results were expressed as mean ± SD. Normality, homogeneity and sphericity of data variances were tested using the Shapiro-Wilk, Levene’s and Mauchly’s tests, respectively. A two-way ANOVA with repeated measures was used to reveal the effects of placebo vs. ω-3 and pre-suppl vs. post-suppl differences between the groups. The magnitudes of differences between the pre- and post-intervention results were expressed as a standardized mean difference (Cohen effect sizes). The criteria to interpret the magnitude of the effect sizes were: <0.2 trivial, 0.2–0.6 small, 0.6–1.2 moderate, 1.2–2.0 large and >2.0 very large. Pearson correlation coefficients were analyzed to determine the inter-variable relationships. The differences were considered significant at *p* < 0.05.

## 3. Results

### 3.1. Somatic and Physiological Characteristics of the Subjects Prior to the Intervention Protocols

There were no significant differences between the somatic (body mass, body height, BMI, FAT, MM, TBW) and physiological (VO2max, HR max) variables of the studied groups (ω-3 suppl. vs. placebo) (Table 1). 

### 3.2. Impact of ω-3 PUFA and Vitamin E and D on the Omega-3 Index 

The ω-3 index and the whole blood compositions of EPA, DHA and AA are presented in Table 3. Before supplementation, the mean value of ω-3 index was 3.9 ± 0.5% (Interpretation of Omega-3 index: high cardiovascular risk: <4.0%; intermediate cardiovascular risk: 4.0–8.0%; and low cardiovascular risk: >8.0%). The pre-suppl AA/EPA ratio was 17.7 ± 6.5. In ω-3 PUFA group, a significant effect of diet on ω-3 index (F = 18.8 *p* < 0.001) and AA/EPA ratio (F = 44.4 *p* < 0.000) was observed. The ω-3 index increased (3.9 ± 0.5 vs. 4.8 ± 0.8, *p* < 0.001) and AA/EPA ratio decreased significantly (17.7 ± 6.5 vs. 8.1 ± 2.4, *p* < 0.001), in response to the supplementation protocol. The post intervention ω-3 index was significantly higher and AA/EPA significantly lower in the ω-3 PUFA compared to the placebo group (*p* < 0.01, *p* < 0.05, respectively). No significant differences were observed between the pre-intervention %AA in both groups (6.6 ± 1.1 vs. 7.3 ± 1.0, *p* = 0.47). The post-intervention %AA was significantly lower in the ω-3 PUFA, compared to the placebo group (*p* < 0.05).

### 3.3. Effect of ω-3 PUFA and Vitamin E and D on Cytokines 

The effects of ω-3 PUFA supplementation and placebo administration on serum adipocytokines and proinflammatory cytokines concentrations in runners were compared after three weeks of each treatment protocol (Table 4). Analysis of variance revealed a significant effect of ω-3 PUFA supplementation on serum adiponectin concentration (F = 22.7; *p* < 0.001) (Table 5). Adiponectin increased significantly (23.8 ± 8.5 vs. 31.4 ± 7.7 µg/mL; *p* < 0.01) in response to ω-3 PUFA dietary protocol. 

ANOVA showed a significant effect of the intervention (pre- vs. post-supplementation) on serum leptin level (F = 4.3; *p* < 0.01). A significantly lower post-suppl serum leptin level compared to pre-suppl (3.3 ± 1.9 vs. 2.7 ± 1.0 ng/dL, *p* < 0.01) was observed. ANOVA revealed a significant effect of the intervention (F = 3.6; *p* < 0.5) on serum NRG-4 level. Ω-3 PUFA supplementation significantly increased the baseline serum level of NRG4 in ω-3 PUFA group (*p* < 0.05).

ANOVA revealed a significant effect of ω-3 PUFA diet on TNFα levels (F = 4.7; *p* < 0.05). A significantly lower post-suppl vs. pre-suppl TNFα level was observed in ω-3 PUFA group (9.7 ± 2.7 vs. 5.6 ± 2.6; *p* < 0.05). Furthermore, post-suppl TNFα level was significantly lower in the suppl vs. placebo group (*p* < 0.05).

### 3.4. Effect of ω-3 PUFA and Vitamin E and D on Antioxidant Status and Lipid Profile

Prior to the intervention protocols, serum α-tocopherol level was below 12 μmol/L, indicating α-tocopherol deficiency [61] in both groups. ANOVA revealed a significant effect of ω-3 PUFA supplementation on α-tocopherol (F = 13.4; *p* < 0.01). Supplementation with ω-3 PUFA significantly increased serum α-tocopherol level (6.5 ± 2.0 vs. 8.7 ± 3.8 µmol/L; *p* < 0.05). The post-intervention α-tocopherol level was significantly higher in the ω-3 PUFA compared to the placebo group (*p* < 0.05) (Table 6). A significant effect of ω-3 PUFA supplementation on MDA levels was also observed (F = 46.1; *p* < 0.000). MDA increased significantly in response to the ω-3 PUFA supplementation (*p* < 0.05) (Table 5). 

A two-way repeated measure ANOVA revealed a significant interaction effect of diet and group on SOD (F = 13.0; *p* < 0.01), GSH (F = 11.4; *p* < 0.01) and GPx (F = 8.9; *p* < 0.05), but not on CAT. A significantly higher SOD activity was observed in response to ω-3 PUFA diet compared to the pre-intervention (1430.8 ± 220.2 vs. 1535.0 ± 259.3 U/gHb; *p* < 0.05] but post-suppl SOD activity was significantly lower compared to the post-placebo (*p* < 0.05). 

ANOVA revealed a significant effect of ω-3 PUFA on serum HDL-C levels (F = 11.1 *p* < 0.01). Significantly higher post-suppl levels of HDL-C compared to pre-intervention (*p* < 0.05) were found in the ω-3 PUFA group. TG levels tended to decrease in response to the intervention in both groups (*p* = 0.05).

### 3.5. Correlations between Omega-3 Index, AA/EPA Ratio and Serum Variables of Adipocytes and Inflammation Markers

A significant negative correlation between the post-suppl ω-3 index and post-suppl AA/EPA ratio (*r* = −0.65; *p* < 0.05) was observed (Figure 1). A significant positive correlation was found between post-suppl ω-3 index and post-suppl NRG4 levels (*r* = 0.7; *p* < 0.05) (Figure 2). An inverse correlation between post-suppl AA/EPO and NRG-4 levels (*r* = −0.65; *p* < 0.05) (Figure 3) was also revealed. We found no evidence of significant association between the ω-3 index and serum adiponectin, leptin and variables of serum lipid profile. A potential relationship between the ω-3 index and AA/EPA ratio and activities of antioxidant enzymes were investigated in the supplemented group. A significant positive correlation was found between the ω-3 index and GPx activity (*r* = 0.62; *p* < 0.05) and the ω-3 index and CAT activity (*r* = 0.62; *p* < 0.05) in the ω-3 PUFA group. The AA/EPA ratio was inversely associated with SOD activity (*r* = −67; *p* < 0.05). A significant increase of the serum adiponectin levels was observed in the supplemented subjects who had greater serum HDL-C levels (*r* = 0.60; *p* < 0.05) (Figure 4). A significant positive correlation between adiponectin levels and SOD activity, as well as a significant negative correlation between TNFα and CAT activity, were detected in ω-3 PUFA group (*r* = 0.78; *p* < 0.001, *r* = −0.72; *p* = 0.01, respectively). 

## 4. Discussion

The present study was undertaken to investigate whether a diet enriched with ω-3 fatty acids, i.e., EPA and DHA, with an addition of vitamin E and D, will have a beneficial effect on the erythrocyte membrane fatty acid concentrations in non-elite marathon runners. It also aimed to determine if this effect will be accompanied with changes in serum levels of adipocyte biomarkers (adiponectin, leptin, NRG-4), proinflammatory cytokines and blood antioxidant status. 

Our results demonstrate that a three-week ω-3 PUFA supplementation (EPA, DHA, vitamin E and D) caused an elevation in the ω-3 index, expressed as a percentage of EPA and DHA in erythrocyte membrane. The ω-3 PUFA supplementation also significantly reduced the AA/EPA ratio, and had a beneficial effect on serum adipocytokines levels compared to the pre-supplementation levels and placebo control group. Baseline adiponectin levels increased and leptin levels decreased in response to the ω-3 PUFA supplementation. We also found significant correlations between the ω-3 index and AA/EPA ratio and serum NRG-4 levels in subjects supplemented with ω-3 PUFA. Serum NRG-4 concentration increased with increases of ω-3 index and EPA + DHA content in erythrocyte membrane. The observed increase of NRG4 in ω-3 PUFA group was evident at the lowest AA/EPA ratio. 

These findings indicate that three weeks of ω-3 PUFA supplementation were sufficient as previously reported [33] to induce beneficial changes in fatty acid composition of RBC and metabolic response. These observations are of particular importance, since the pre-intervention RBC ω-3 content was low in the examined group of runners, suggesting a high cardiovascular risk (ω-3 index < 4.0%) [1,53]. The ω-3 index increased significantly in response to 3-week supplementation (3.9 ± 0.5 vs. 4.8 ± 0.8%), reducing the cardiovascular risk, although it was still below a target range of >8.0% [1,57]. It can be expected that a longer duration of ω-3 supplementation could lead to further increases of the ω-3 index. Studies have shown that the incorporation of EPA + DHA into membranes of RBC seems to be more effective in those presenting low baseline levels of EPA + DHA. Moreover, the incorporation of EPA + DHA can be regulated to some degree until reaching a saturation point [62].

Low ω-3 index has been previously reported in a population of endurance-trained athletes [1,28,57]. It has been also demonstrated that the ω-3 index decreases gradually in response to a greater weekly running distance [29] and higher training intensity [32]. The ω-3 PUFA-induced positive changes in arterial compliance and endothelial function, as well as increased lipid oxidation and decreased lipogenesis, could exert a cardio-protective role in endurance athletes [63]. These findings emphasize the need to recognize the importance of an optimal EPA/DHA intake in athletes. To date, the recommended daily allowance for EPA and DHA has not yet been clearly defined, although certain recommendations exist [62]. The World Health Organization established a need for consumption of 250–500 mg/day of EPA + DHA [64]. Cardioprotective effects seem to be achieved at higher doses of 1 to 3–5 g/day of EPA + DHA [65], and in sport activities a similar dose has been recommended [62,66].

Data on the role of ω-3 PUFA supplementation in optimizing inflammatory responses, the processes of tissue restoration and cardiovascular health in athletes are still sparse. Bloomer et al. [67] demonstrated that 2224 mg EPA + 2208 mg DHA supplementation (daily EPA + DHA intake of ~4.4. g) for six weeks resulted in a significant increase in blood levels of both EPA and DHA in exercise-trained men. The ω-3 PUFA diet decreased resting levels of inflammatory biomarkers but had little effect on exercise-induced inflammation or oxidative stress [67]. In the present study, the members of ω-3 PUFA group were daily supplemented with lower daily doses of EPA + DHA (852 mg EPA + 1602 mg DHA per day). Nevertheless, the daily intake of ~2.5 g of EPA + DHA induced beneficial metabolic effects. Similar total daily doses of EPA + DHA were used in a study by McAnulty et al. [27] who demonstrated that 2000 mg EPA + 400 mg DHA significantly increased plasma EPA and DHA levels in a group of cyclists, but also led to a significant rise of markers of oxidative stress (F2-isoprostanes) after exhaustive exercise. Moreover, the authors [27] demonstrated that antioxidant supplementation (combination of vitamin E, C and selenium) modestly attenuated the ω-3 fatty acid associated increase in the markers of oxidative stress. This finding suggests that the increased EPA + DHA intake should be accompanied with a greater consumption of antioxidants, especially α-tocopherol (vitamin E), which serves as a free radical scavenger in lipid components of the cell and plasma lipoproteins [68]. The antioxidant function of α-tocopherol is critical for the prevention of oxidation of tissue EPA + DHA [69]. In the present study, the baseline serum α-tocopherol levels were low in both groups, but increased significantly (6.5± 2.0 vs. 8.7 ± 3.8 μmol/L) in response to the administration of the ω-3 supplements which contained 12 mg of α-tocopherol per capsule. Nevertheless, the post-suppl plasma α-tocopherol levels were still below the recommended range ~30 μmol/L [61,70], demonstrating its probable insufficient dietary intake. The assessment of mean dietary intake of α-tocopherol conducted before the supplementation protocol demonstrated that subjects from both groups were not meeting the daily recommended allowance (RDS) of 15 mg [61]. Nevertheless, the daily supplementation of 72 mg (12 mg × six capsules/day) of α-tocopherol would seem to be sufficient to increase its plasma levels to a greater degree than observed. This confirms that an increased dietary intake of PUFA decreases plasma α-tocopherol [70]. Therefore, the vitamin E requirement increases with a rise in PUFA consumption. 

We found that intake of 2454 mg of EPA + DHA/day for three weeks also resulted in a significant increase in antioxidant enzymatic (SOD and GPx) defense compared to pre-supplementation levels and the control group. A greater SOD and GPx expression in response to the ω-3 PUFA diet positively correlated with higher levels of a lipid peroxidation marker, MDA. It has been previously reported that a diet rich in ω-3 PUFA may increase lipid peroxidation, and therefore higher ROS production can be expected [5,27]. Our findings suggest that oxidative stress induced by an increased fatty acid consumption may be responsible for the increased antioxidant enzymatic and non-enzymatic defense. Nevertheless, an optimal intake of ω-3 PUFA and potentially α-tocopherol to upregulate endogenous antioxidant defense mechanism are yet to be defined in the athletic population [71,72]. Additionally, exercise training load should be taken into account when defining such intakes [73]. 

The AA/EPA ratio can be used to better evaluate inflammation and lipid content of cell membranes [74]. AA and EPA are both incorporated into cell membrane when AA/EPA ratio is between 1:1 to 5–10:1 [75]. When the ratio increases, however, the incorporation of AA is preferred, leading to increased production of the 2-series eicosanoids and inflammation [76]. In the present study, the baseline AA/EPA ratio was elevated in both groups, but decreased significantly in response to ω-3 PUFA supplementation (17.7 ± 6.5 vs. 8.1 ± 2.4). The reduction in AA/EPA ratio resulted from an increased in response to ω-3 supplementation %EPA (*p* < 0.001) as well as a tendency to lower %AA (6.6 ± 1.1 vs. 6.1 ± 1.2). It is worth noting here that even a small decrease in %AA could reduce inflammation [15,16]. The post intervention %AA was significantly lower in the ω-3 PUFA compared to the placebo group, but nevertheless, this finding should be interpreted with caution as a tendency for elevated pre-intervention AA values were observed in the placebo group. Since the study subjects’ diet was controlled and none of them consumed fish or other oils that would additionally modify the dietary fatty acids intake, our findings suggest that ω-3 supplementation could have a beneficial effect on both ω-3 (increase) and ω-6 (decrease) content in the biological membranes. 

Our study findings also show that 3-week supplementation with ω-3 PUFA significantly increased serum adiponectin and NRG-4 levels, and reduced serum leptin levels. Moreover, the increased adipocytokines, i.e., adiponectin and NRG-4 production seemed to have a positive effect on blood lipid profile, inflammation and antioxidant defense. Similarly, a recent systematic review and meta-analysis involving 685 individuals with prediabetes and Type 2 diabetes revealed that ω-3 PUFA supplementation for a period of 12–24 weeks increased levels of adiponectin and reduced levels of a proinflammatory cytokine, TNFα [77]. Interestingly, ω-3 PUFA supplementation of a shorter duration (<12 weeks) was more effective in reducing TNFα compared to a duration of >12 weeks [77]. However, the authors concluded that due overall study heterogeneity and potential publication bias their results should be interpreted with caution [77].

In the present study, a significant effect of ω-3 PUFA on serum adiponectin concentrations was associated with lower TNFα and leptin levels, as well as increased HDL-C concentrations. Adiponectin has been shown to exert important physiological properties with a profound impact on glucose and fatty acid metabolism [36,78]. A strong positive correlation between serum adiponectin concentration, insulin sensitivity and increased glucose transport in skeletal muscle has been previously reported [35,79]. It has been shown that adiponectin not only increases skeletal muscles’ glucose uptake, maintaining normoglycaemia, but also increases the oxidation of fatty acids via AMP kinase activation pathway, leading to a decrease in plasma-free fatty acids levels [34,35]. Therefore, increasing levels of adiponectin may be a promising strategy for improving metabolic and cardiovascular function. Further research is needed to explain the mechanism of its regulation and function in healthy individuals subjected to intense and prolonged exercise. 

The observed significant reduction of TNFα concentration in response to ω-3 PUFA is of great importance, since TNFα is one of the main proinflammatory cytokines, also secreted by the adipose tissue, which chronically elevated levels contribute to long-term inflammation and lead to a range of health complications [80]. It should be noted, that not only EPA + DHA but also vitamin E and vitamin D could have contributed to the reduction of TNFα levels in the supplemented group. Both vitamin D [81] and vitamin E [82] have been shown to decrease the expression of proinflammatory cytokines such as TNFα. 

In the present study, the reduced serum TNFα levels were associated with a decreased concentration of leptin. Leptin is produced by adipocytes and its plasma concentration rises with increases of fat mass, insulin and TNFα levels [40]. Other factors, such as a low-fat diet, free fatty acids and growth hormone, have been shown to decrease leptin secretion into the bloodstream [36,78]. Our results are in line with previously reported data showing reduction of serum leptin level in response to ω-3 PUFA supplementation [10,83,84].

A significant increase of HDL-C concentration and a tendency to lower LDL-C was observed in the ω-3 PUFA group. Interestingly, TG concentration decreased in both studied groups, suggesting that ω-3 PUFA supplementation was not associated with this change. Nevertheless, since elevated LDL-C and reduced HDL-C concentrations are a known risk factor for development of CVD [85], their improved concentration in response to the supplementation is an important finding. 

It has been recently reported that ω-3 fatty acids stimulate adipose tissue metabolism via activation of brown and beige adipose tissue, which play a crucial role in maintaining energy homeostasis through non-shivering thermogenesis [86,87]. Consequently, brown and beige adipose tissue release factors which mediate lipid metabolism and thermogenic activation [43,47,88]. One of these factors is NRG-4 [51], a member of the neuregulins family of ligands, which have been shown to regulate different aspects of glucose and lipid metabolism [48]. NRG-4 reduces gluconeogenesis and lipogenesis, and increases fatty acid oxidation [43,88]. A recent study showed that brown adipose tissue-derived NRG-4 may also promote the growth of neurites in adipose tissue, increasing sympathetic innervation, stimulating brown adipose tissue activity and browning of white adipose tissue [46,51]. In our study, serum levels of NRG-4 increased in response to ω-3 PUFA diet. Furthermore, NRG-4 levels positively correlated with ω-3 index, while as expected, lower AA/EPO ratio was associated with greater NRG-4 secretion. These findings indicate a possible role of NRG4 in the white adipose tissue browning, as well as possible increased lipid metabolism [46,47,51]. The benefits of NRG4 in response to ω-3 PUFA diet should be treated with caution, however, as no analyses of NRG-4 effect on brown adipocyte tissue thermogenesis were performed in this study. 

To summarize, three weeks of ω-3 PUFA extract with added vitamin E and D supplementation had a beneficial effect on the erythrocyte fatty acids composition of the non-elite long-distance runners. The supplementation modulated plasma concentration of adiponectin, leptin and NRG-4. Moreover, the increased adipocytokines, i.e., adiponectin and NRG-4 production seemed to have a positive effect on lipid profile, markers of inflammation and antioxidant defense, potentially counteracting the PUFA consumption-induced lipid peroxidation. 

The observed modifications may contribute to the prevention of cardiovascular events in the studied runners [7,8,21,35,40]. This appears to be particularly important, as individuals with a low ω-3 index are at increased risk for chronic inflammation diseases [89], myocardial infarction and cardiac death [1,3,26]. Moreover, the positive correlation between the ω-3 index, NRG4 levels and markers of antioxidant defense may be associated with increased ability to reduce fatigue and with prevention of cardiovascular events after high-intensity exercise. The possible mechanisms of exercise-induced adaptive mechanisms in response to the ω-3 PUFA diet were not a focus of this paper. 

### Limitations

There are some limitations of the study. The study included a small group of recreational marathon runners (*n* = 24) who consumed either the placebo or the ω-3 PUFA diet for three weeks. Furthermore, not only an effect of omega 3 was studied but of vitamin E and D added to the supplement. Further investigations with a larger sample, a longer supplementation period, and a control group supplemented with omega 6 oil plus vitamin E and D could provide more specific results regarding omega 3 effects. It could also enable a generalization of the study. These preliminary findings highlight the requirement for further studies into optimal ω-3 PUFA intake but also an effect of vitamin E and vitamin D in maximizing physical performance and more importantly, preventing morbidity and mortality in the population of elite and non-elite athletes.

## 5. Conclusions

Three weeks of ω-3 PUFA extract with added vitamin E and D supplementation had a positive effect on lipid composition of erythrocytes, serum adiponectin, leptin and NRG4 levels, and caused a significant decrease in post-exercise proinflammatory cytokine levels in recreational runners. We concluded that ω-3 PUFA supplementation may play an important role in the improvement of adipocyte tissue function and prevention of cardiovascular complications in recreational runners.

## Figures and Tables

**Figure 1 nutrients-13-00456-f001:**
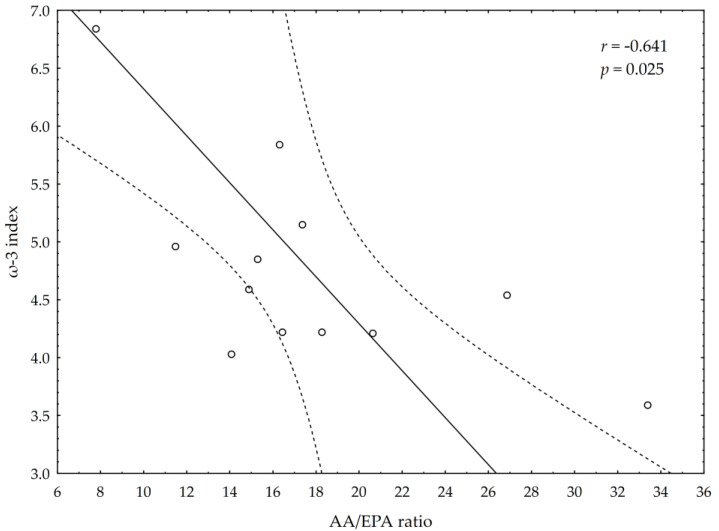
Correlation between omega-3 index (ω-3 index) and AA/EPA ratio.

**Figure 2 nutrients-13-00456-f002:**
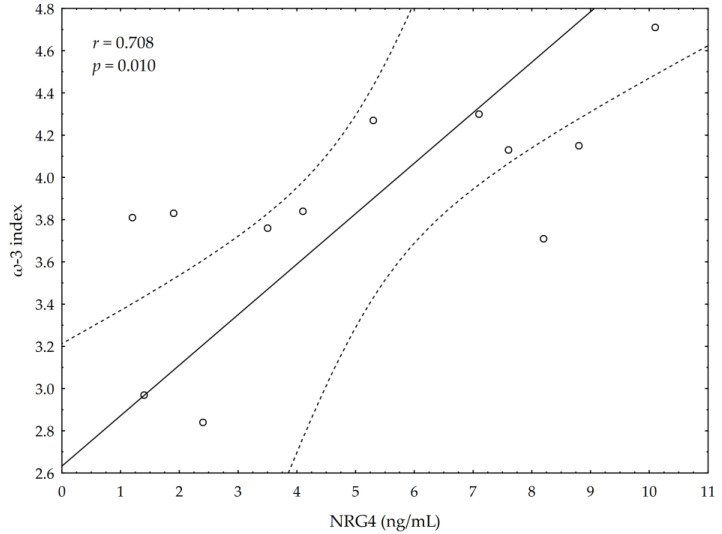
Correlations between omega-3 index (ω-3 index) and serum neuregulin 4 (NRG4) levels.

**Figure 3 nutrients-13-00456-f003:**
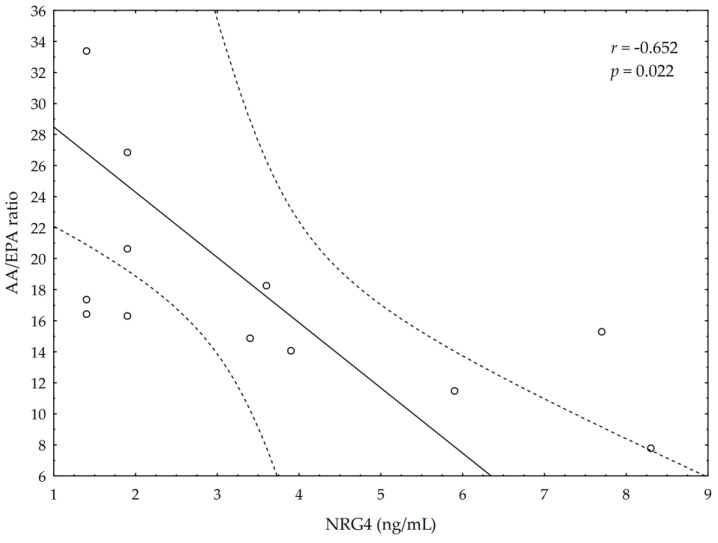
Correlation between AA/EPA ratio and serum neuregulin 4 (NRG4) levels.

**Figure 4 nutrients-13-00456-f004:**
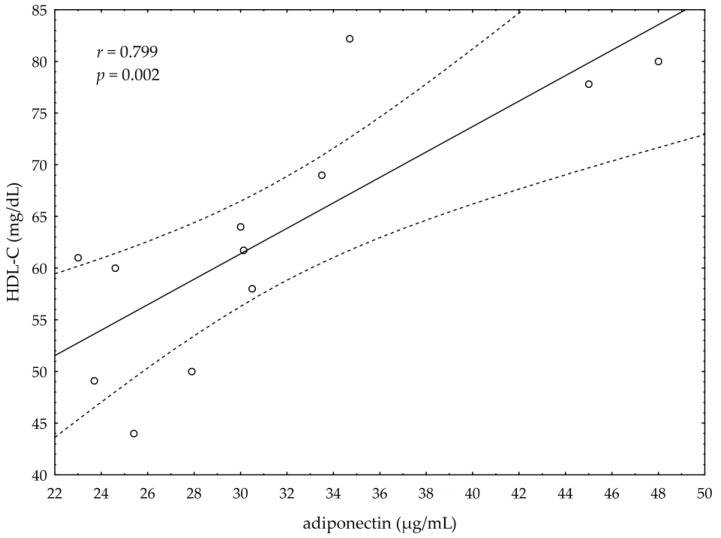
Correlation of high density lipoprotein cholesterol (HDL-C) and serum adiponectin levels.

**Table 1 nutrients-13-00456-t001:** Participants’ characteristics (mean, SD).

Variables	(ω-3 PUFA *n* = 12)	Placebo (*n* = 12)
Age (years)	33.3 ± 6.8	35.0 ± 5.8
Body mass (kg)	74.3 ± 9.7	74.4 ± 7.6
Body height (cm)	177.2 ± 5.2	178.1 ± 6.8
BMI (kg/m^2^)	23.5 ± 1.9	23.5 ± 2.0
FAT (%)	13.6 ± 3.3	13.5 ± 4.1
MM (kg)	36.8 ± 5.3	36.9 ± 3.5
TBW (L)	47.2 ± 6.4	47.5 ± 5.4
VO_2_max (mL/kg/min)	55.0 ± 9.2	58.1 ± 7.4
HR max (b/min)	182.0 ± 10.0	186.0 ± 10.0
Training volume (min/week)	360.0 ± 55.0	380.0 ± 60.0

BMI—body mass index, FAT—percent of body fat, MM—skeletal muscle mass, TBW—total body water, VO_2_max—maximal oxygen uptake, HR max—heart rate maximum.

**Table 2 nutrients-13-00456-t002:** Mean daily energy, fat, carbohydrate, protein, vitamins and fatty acids intake in the supplemented (ω-3 PUFA) and placebo group (Placebo) (mean, SD).

Variables	ω-3 PUFA (*n* = 12)	Placebo (*n* = 12)
Energy [kcal/kg/day]	29.6 ± 3.0	28.0 ± 2.0
Fat [%]	31.7 ± 9.6	30.8 ± 8.3
Carbohydrate [%]	46.1 ± 6.6	46.7 ± 8.5
Protein [%]	22.8 ± 5.4	22.4 ± 3.3
Vitamin C [mg/day]	171.2 ± 73.0	176.4 ± 122.0
Vitamin E [mg/day]	11.8 ± 3.6	11.2 ± 4.7
Vitamin D [µg/day]	7.9 ± 5.0	7.8 ± 7.3
Cholesterol [mg/day]	436.0 ± 228.6	423.1 ± 159.0
SFA [g/day]	37.4 ± 15.2	33.5 ± 9.6
MUFA [g/day]	29.6 ± 10.1	26.0 ± 8.7

SFA—saturated fatty acids; MUFA—monounsaturated fatty acids.

**Table 3 nutrients-13-00456-t003:** Omega-3 Index and other fatty acids content in the erythrocyte membranes in the supplemented group (ω-3 PUFA) and placebo group (Placebo) (mean, SD).

Variables	ω-3 PUFA (*n* = 12)		*Placebo* (*n* = 12)		*p*
	Pre-Suppl	Post-Suppl	Pre-Placebo	Post-Placebo	Post-Suppl vs. Post-Placebo
Omega-3 Index [%]	3.9 ± 0.5	4.8 ± 0.8 ***	3.5 ± 0.3	3.9 ± 0.4	0.01
ω-3 FA [%]	3.9 ± 0.5	4.8 ± 0.9 ***	3.5 ± 0.4	4.1 ± 0.5	0.05
Alpha-Linolenic [%]	0.1 ± 0.1	0.1 ± 0.1	0.1 ± 0.0	0.2 ± 0.0	ns
EPA [%]	0.4 ± 0.2	0.8 ± 0.3 ***	0.5 ± 0.1	0.7 ± 0.1	ns
DPA [%]	0.9 ± 0.2	1.0 ± 0.1	0.9 ± 0.2	1.0 ± 0.2	ns
DHA [%]	2.5 ± 0.6	2.9 ± 0.6 **	2.0 ± 0.3	2.1 ± 0.4	0.05
AA/EPA	17.7 ± 6.5	8.1 ± 2.4 ***	14.2 ± 3.4	12.0 ± 3.7	0.05
ω-6 FA [%]	28.7 ± 2.8	27.4 ± 2.4	28.8 ± 3.1	28.9 ± 3.5	ns
AA [%]	6.6 ± 1.1	6.1 ± 1.2	7.3 ± 1.0	7.5 ± 1.0	0.05
SFA [%]	45.6 ± 1.8	47.3 ± 1.3	45.0 ± 1.9	45.6 ± 2.0	ns
MUFA [%]	20.8 ± 1.9	20.1 ± 1.7	22.7 ± 2.5	21.0 ± 2.1	ns
SFA/MUFA	2.2 ± 0.2	2.4 ± 0.2	2.0 ± 0.2	2.2 ± 0.2	ns
TRANS index [%]	0.2 ± 0.1	0.3 ± 0.1	0.3 ± 0.1	0.3 ± 0.1	ns

Omega-3 index—sum of EPA + DHA expressed as % of total weight of fatty acids in erythrocyte membrane; ω-3 FA—sum of omega 3 fatty acids; EPA—eicosapentaenoic acid; DPA-docosapentaenoic acid; DHA—docosahexaenoic acid; AA/EPA—arachidonic acid to eicosapentaenoic acid ratio; ω-6 FA—sum of omega 6 fatty acids; AA—arachidonic acid; SFA—saturated fatty acids; MUFA—monounsaturated fatty acids; SFA/MUFA—saturated fatty acids to monounsaturated fatty acids ratio; TRANS index—concentration of trans fatty acids. * Significant difference: ** *p* < 0.01; *** *p* < 0.001. All variables apart from AA/EPA and SFA/MUFA are expressed as % of total weight of fatty acids in erythrocyte membrane.

**Table 4 nutrients-13-00456-t004:** Adipocytokines, cytokines and lipid serum concentrations in the supplemented group (ω-3 PUFA) and placebo group (Placebo) (mean, SD).

Variables	ω-3 PUFA (*n* = 12)		Placebo (*n* = 12)		*p*
	Pre-Suppl	Post-Suppl	Pre-Placebo	Post-Placebo	Post-Suppl vs. Post-Placebo
Adiponectin [µg/mL]	23.8 ± 8.5	31.4 ± 7.7 **	28.8 ± 8.5	30.1 ± 13.5	ns
Leptin [ng/mL]	3.3 ± 1.9	2.7 ± 1.0 **	2.6 ± 0.4	2.7 ± 0.3	ns
NRG-4 [ng/mL]	3.6 ± 2.1	5.4 ± 2.1 *	2.9 ± 2.0	3.2 ± 1.6	ns
TNFα [pg/mL]	9.7 ± 2.7	5.6 ± 2.6 *	13.7 ± 7.4	12.5 ± 2.4	0.05
IL-6 [pg/mL]	1.4 ± 1.3	1.9± 1.8	1.5 ± 1.3	2.2 ± 2.0	ns
TCh [mg/dL]	197.0 ± 31.4	210.0 ± 34.7	184.7 ± 15.4	187.3 ± 11.7	ns
HDL-C [mg/dL]	57.6 ± 12.3	68.9 ± 13.0 *	60.5 ± 5.6	68.2 ± 11.8	ns
LDL-C [mg/dL]	114.2 ± 27.1	106.8 ± 30.3	104.0 ± 21.7	98.4 ± 15.2	ns
TG [mg/dL]	118.0 ± 27.1	103.0 ± 25.4	113.5 ± 25.1	87.7 ± 16.3	ns
BG [mg/dL]	80.3 ± 11.0	85.7 ± 12.0	90.2 ± 10.4	92.0 ± 15.1	ns

NRG4—Neuregulin-4; TNFα—tumor necrosis factor alpha; IL-6—interleukin 6; TCh—total cholesterol; HDL—C-high density lipoprotein cholesterol; LDL-C—low density lipoprotein cholesterol; TG—triglycerides; BG—blood glucose. * Significant difference: * *p* < 0.05; ** *p* < 0.01.

**Table 5 nutrients-13-00456-t005:** Effect of ω-3 PUFA supplementation on adipocytokines, cytokines, lipid profile and antioxidant enzymes levels in supplemented group (ω-3 PUFA).

Variable	F	*p*	ɳ^2^	α
Adiponectin [µg/mL]	22.7	0.000	0.51	1.0
Leptin [ng/mL]	2.3	0.14	0.09	0.3
NRG4 [ng/mL]	2.2	0.17	0.26	0.3
TNF α [pg/mL]	4.7	0.05	1.18	0.6
IL-6 [pg/mL]	0.4	0.59	0.02	0.2
Cholesterol [mg/dl]	0.7	0.42	0.03	0.1
HDL [mg/dL]	11.1	0.003	0.34	0.9
LDL [mg/dL]	0.4	0.53	0.02	0.1
TG [mg/dL]	1.2	0.31	0.05	0.3
MDA [µmoL/L]	46.1	0.000	0.68	1.0
SOD [U/gHb]	13.0	0.002	0.37	0.9
GPx [U/gHb]	8.9	0.007	0.29	0.8
CAT [U/gHb]	0.9	0.35	0.04	0.2
GSH [µg/gHb]	11.4	0.002	0.34	0.9
α-Tocopherol [µmoL/L]	13.4	0.004	0.55	0.9

F-Fisher ratio, *p*-value, η^2^-eta squared, and α-effect size, MDA—malondialdehyde; SOD—superoxide dismutase, GPx—glutathione peroxidase; CAT—catalase; GSH—reduced glutathione.

**Table 6 nutrients-13-00456-t006:** Oxidative status in the supplemented group (ω-3 PUFA) and placebo group (Placebo) (mean, SD).

Variables	ω-3 PUFA (*n* = 12)		Placebo (*n* = 12)		*p*
	Pre-Suppl	Post-Suppl	Pre-Placebo	Post-Placebo	Post-Suppl vs. Post-Placebo
MDA [µmol/L]	5.0 ± 1.1	5.6 ± 1.3 *	5.3 ± 0.8	6.3 ± 1.3	ns
SOD [U/gHb]	1430 ± 220	1535 ± 259 **	1330 ± 115	1620 ± 407	0.05
GPx [U/gHb]	45.1 ± 9.4	57.4 ± 7.6 **	49.4 ± 11.4	59.5 ± 15.4	ns
CAT [U/gHb]	179.6 ± 23.4	185.6 ± 34.1	212.5 ± 26.0	207.4 ± 31.6	ns
GSH [µg/gHb]	2.7 ± 0.3	2.5 ± 0.3	2.8 ± 0.4	2.6 ± 0.2	ns
α-Tocopherol [µmol/L]	6.5 ± 2.0	8.7 ± 3.8 *	4.8 ± 1.7	4.9 ± 2.5	0.05

* Significant difference: * *p* < 0.05; ** *p* < 0.01.

## Data Availability

Upon request from the first author.

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
