# Peer review of "The Effect of Omega-3 Fatty Acid Supplementation on Serum Adipocytokines, Lipid Profile and Biochemical Markers of Inflammation in Recreational Runners"

_nutrients, 2021, doi:10.3390/nu13020456_

Round 1

Reviewer 1 Report

The authors investigated the effect of an omega-3 PUFA supplementation on various biochemical parameters related to CVD, blood antioxidant defense, and others in blood serum. The study seems to be well designed and executed. The authors carefully explained the results and all conclusions made were supported by the data. However, I have a few comments for the author that should be explained.

  1. Did you take into account when had the participants completed the marathon run before entering the study? Did you consider the time in which the participants finished the marathon, or whether they finished the run at all?
  2. Why did you choose FAs supplementation in a form of free FAs and not, for instance, TAGs or phospholipids?
  3. Could you be more specific regarding the FAs determination performed by the Italian laboratory? Do I understand correctly that they used some sort of biochemical kit for the specific determination of a particular FAs ratio (AA/EPA)?

Minor comments:

  • Line180: unify the number of decimal places
  • Table3: Specify the units of the variables. %?
  • Line242: A dot for the sentence ending is missing between ,,observed,, and ,,ANOVA,,
  • Table5: Define the calculated statistical parameters (F means Fisher ratio?, etc.); Table caption: I don´t see any ,,mean,, and ,,SD,, in the table; Move the full names of the variables from the Table 6 to the Table 5 because they are mentioned first there.
  • Table6: The number after the decimal is not necessary for the variable SOD.
  • Line351: suggests
  • Line386: delete one ,,that,,

Author Response

Responses to Reviewer1

We wish to thank  the Reviewer for the thorough review of the manuscript and valuable comments. We have addressed the issues indicated in the review report. We believe that the revised version of the manuscript will  meet the journal’s publication requirements.

  1. Did you take into account when had the participants completed the marathon run before entering the study? Did you consider the time in which the participants finished the marathon, or whether they finished the run at all?

Answer: The description of the study group was changed in accordance with the Reviewer’s comments. We have added that the study participants completed at least one marathon  in the last twelve months and that it was  an important inclusion criterion. The marathon finish times varied between 3 hours 23 minutes and 4 hours 18 minutes, which has been added to  the Methods section.

  1. Why did you choose FAs supplementation in a form of free FAs and not, for instance, TAGs or phospholipids?

Answer: Thank you for the comment. We have explained that the previous studies reported   health benefits of omega-3 PUFA which may be attributed to the biological effects on inflammation response, lipoprotein metabolism, and mitochondrial function. Moreover, omega-3 PUFAs have been shown  to lower the concentration of triglycerides, which at high levels contribute to the development of cardiovascular disease. TAGs or phospholipids have a limited effect on fat metabolism and lowering the risk of atherosclerosis compared to the extracts enriched with EPA and DHA (Mozaffarian D, Rimm EB. Fish intake, contaminants, and human health: evaluating the risks and the benefits. JAMA, 2006; 296: 1885–1899.; Harris WS, Bulchandani D. Why do omega-3 fatty acids lower serum triglycerides? Curr Opin Lipidol, 2006; 17: 387–393.; Park Y, Harris WS. Omega-3 fatty acid supplementation accelerates chylomicron triglyceride clearance. J Lipid Res, 2003; 44: 455–463).

  1. Could you be more specific regarding the FAs determination performed by the Italian laboratory? Do I understand correctly that they used some sort of biochemical kit for the specific determination of a particular FAs ratio (AA/EPA)?

Answer: As suggested by the reviewer, we have added a more specific description of the FAs determination: the evaluation of % PUFA (omega 3: EPA, DHA/omega 6:LA, AA)/monounsaturated (MUFA)/saturated fatty acid (SFA)) and TRANS fatty acids in erythrocytes cell membrane was measured. To determine the omega index, five finger blood spots were taken and placed on a special spot card. Omega index was determined using GC-FID Gas Chromatography-Flame Ionization Detector. The Omega Index measured in dried blood spots (DBS) is a qualitative (%) not quantitative (µg/mL) expression of total fatty acids. DBS analysis used is a  repeatable measurement which is relatively non-invasive.

Minor comments:

  • Line180: unify the number of decimal places
  • Table3: Specify the units of the variables. %?
  • Line242: A dot for the sentence ending is missing between ,,observed,, and ,,ANOVA,,
  • Table5: Define the calculated statistical parameters (F means Fisher ratio?, etc.); Table caption: I don´t see any ,,mean,, and ,,SD,, in the table; Move the full names of the variables from the Table 6 to the Table 5 because they are mentioned first there.
  • Table6: The number after the decimal is not necessary for the variable SOD.
  • Line351: suggests
  • Line386: delete one ,,that,,

Answer: Thank you for the valuable minor comments. We have corrected the text and tables.

Reviewer 2 Report

The manuscript by Zebrowska et al., provide data to suggest that omega-3 polyunsaturated fatty acid (PUFA) supplementation might play an important role in the improvement of adipocyte tissue function and cardiovascular complications in athletes. The authors enrolled 24 participants to assess potential benefits of EPA/DHA PUFA supplementation on serum adipokine levels. Following 3 weeks of supplementation or not, the authors report an increase in adiponectin, NRG4 levels and a decrease in leptin concentration and TNFa levels. Collectively, these conclusions are supported by the data however some issues deserve attention.

1). The authors conducted a 3 week volunteer study where participants were randomly allocated into two groups. The omega-3 group participants were supplemented with gelatinous capsules containing standardized omega free fatty acid extract, vitamin E and vitamin D. Other than consuming gelatin capsules, placebo composition details are lacking. Details regarding how fatty acid composition was controlled (i.e., EPA/DHA vs corn oil amounts) is lacking. This information is required to effectively evaluate the study results.

2).   Placebo group “pre” values for AA appear to be significantly elevated compared with omega-3 PUFA “pre” and not simply just after supplementation. Clarification of this point is needed as these data may also help to explain the findings of omega-3 supplementation. These data call into question if the benefits of supplementation are directly related to enhanced omega-3 or a decrease in AA levels.

3). Composition details regarding omega-3 supplementation capsules is missing as “free fatty acid extract” may contain more than EPA and DHA alone.

Author Response

Responses to Reviewer 2

We  wish  to  thank  the  Reviewer  for  the  thorough  review  of  the  manuscript.

1). The authors conducted a 3 week volunteer study where participants were randomly allocated into two groups. The omega-3 group participants were supplemented with gelatinous capsules containing standardized omega free fatty acid extract, vitamin E and vitamin D. Other than consuming gelatin capsules, placebo composition details are lacking. Details regarding how fatty acid composition was controlled (i.e., EPA/DHA vs corn oil amounts) is lacking. This information is required to effectively evaluate the study results.

Answer: We agree with the  reviewer’s comment and we have added the exact details regarding placebo composition and a method of fatty acid composition control. We have added  the following text: Placebo capsules contained microcrystalline cellulose, magnesium stearate and lactose monohydrate instead of fish extract. Placebo capsules (Natural Pharmaceutic, Warsaw, Poland) were administered at a dose of six capsules per day, four in the morning and two in the evening, for three weeks.

The intake of fatty acids was controlled. The experimental diet were formulated with food items commonly available. None of the respondents consumed fish or other oils that would additionally modify the content of fatty acids in the diet. In the European diet, corn oil is rather not consumed.  Subjects were asked to consume all recommended foods  and a  compliance with the research diet was evaluated during  obligatory weekly visits in the laboratory. The subjects were then asked whether they had consumed all the recommended foods and if any additional foods had been eaten.

2).   Placebo group “pre” values for AA appear to be significantly elevated compared with omega-3 PUFA “pre” and not simply just after supplementation. Clarification of this point is needed as these data may also help to explain the findings of omega-3 supplementation. These data call into question if the benefits of supplementation are directly related to enhanced omega-3 or a decrease in AA levels.

We would like to thank the Reviewer for this valuable comment. No significant differences were observed between the pre intervention values for %AA in both groups (6.6 ±1.1 vs 7.3 ± 1.0, p=0.47). However, placebo ‘’pre’’ AA levels were elevated and could have led to a significant difference between post- suppl vs post placebo as noted by the reviewer. We have clarified this in the discussion. Also, as pointed out by the reviewer, we have added a section explaining that both enhanced omega 3 and a tendency to lower omega-6 (i.e. AA) in response to omega 3 supplementation could have health benefits.

3). Composition details regarding omega-3 supplementation capsules is missing as “free fatty acid extract” may contain more than EPA and DHA alone.

Answer: This part of work has been changed to include the reviewer's comments: Omega 3 supplementation capsules contained concentrated, refined sardine and anchovy oil (72%) containing 90% of ω-3 PUFA (EPA and DHA), gelatin, humectant (glycerol), DL-alpha-tocopherol, cholecalciferol. Apart from EPA and DHA, it does not contain significant amounts of other fatty acids. The supplement was produced in Norway (Pharmatech AS) for Natural Pharmaceutic, Warsaw, Poland.

Round 2

Reviewer 2 Report

The reviewer appreciates the responsiveness of the authors. The authors should clarify in the methods if the study was performed in a blinded fashion. If not, this should be included in a limitations section in the discussion section along with other limitations of the study.

Author Response

We  wish  to  thank  the  Reviewer  for  the   valuable comments. We have corrected the English language and addressed the issues indicated in the review report.

We believe that  the revised version of the manuscript can meet the journal publication requirements. The following corrections have been implemented (all of them have been marked in blue).

  1. The authors should clarify in the methods if the study was performed in a blinded fashion. If not, this should be included in a limitations section in the discussion section along with other limitations of the study.

Answer: We have added in the methods that the study was performed in a blinded fashion “participants in the study were included in this blind, randomized placebo-controlled trial and assigned to either the placebo or the ω-3 PUFA group“.